# Effects of a school-based intervention on 24-hour movement behaviors in adolescents: A quasi-experimental study

Jaminson Raul Ricardo-Sejin ®*, Carlos Mario Arango-Paternina, Fredy Alonso Patiño-Villada

University Institute of Physical Education and Sports, University of Antioquia, Medellín, Colombia

* jaminson.ricardo@udea.edu.co

## Abstract

Interest in integrating 24-hour movement behaviors (sleep, physical activity, and screen time) has increased in recent years, supporting global guidelines that emphasize their relevance for adolescent health. However, there remains a need for studies that assess the effectiveness of interventions aimed at improving these behaviors. This study aimed to evaluate the effectiveness of a school-based intervention grounded in Self-Determination Theory on: (1) 24-hour movement behaviors; (2) adherence to 24-hour movement behaviors recommendations; (3) health-related quality of life; and (4) Self-Determination Theory constructs associated with 24-hour movement behaviors. A quasi-experimental design with experimental and control groups was conducted, including pre-test, post-test, and follow-up assessments. The study involved 82 adolescents aged 14–17 from two conveniently selected schools in the city of Bello, Colombia. The 12-week intervention included educational, family, and extracurricular components. Effectiveness was analyzed using mixed models and relative risk. The results revealed significant effects of the intervention on screen time (weekdays: $F(4.68, 156)=2$, $p = .01$; weekend: $F(5.51, 156)=2$, $p = .005$; total week: $F(6.32, 156)=2$, $p = .002$) and sleep (weekend: $F(6.09, 156)= 2$, $p = .003$; total week: $F(3.88, 156)=$, $p = .02$). A significant increase in meeting the weekday sleep recommendation was observed in the experimental group (RR: 1.85, IC95% 1.05–3.25), along with improvements in the Self-Determination Theory constructs of competence and relatedness in physical activity. The intervention demonstrated partial effectiveness, achieving improvements in screen time and sleep. This study contributes to the understanding of the design and implementation of educational interventions targeting 24-hour movement behaviors. Future research should refine the integration of motivational strategies to improve physical activity and overall adherence to recommendations, while also exploring the scalability and sustainability of these school-based interventions.

**Data availability statement:** All relevant data are within the manuscript and its Supporting Information files.

**Funding:** The author(s) received no specific funding for this work.

**Competing interests:** The authors have declared that no competing interests exist.

# 1. Introduction

In the current landscape of public health, 24-hour movement behaviors (24-HMB) have emerged as a critical factor in adolescent well-being. This comprehensive concept encompasses the dynamic interplay among physical activity (PA), sedentary behavior (SB), and sleep, recognizing that these components are not independent but mutually influence one another to shape overall health [1,2]. Adolescence is a stage of the life course characterized by physical, psychological, and social changes, offering a unique context to establish behavioral patterns that may persist into adulthood [3]. In this regard, understanding and optimizing 24-HMB during adolescence is essential for the prevention of non-communicable diseases, the promotion of mental health, and the development of an active lifestyle [4].

24-HMB were introduced in 2016 by the Canadian Society for Exercise Physiology as the "Canadian 24-Hour Movement Guidelines" [5], which recommend: (a) at least 60 minutes of daily moderate-to-vigorous physical activity (MVPA); (b) no more than 2 hours of recreational screen time (ST); and (c) uninterrupted sleep of 9–11 hours for children aged 5–13 and 8–10 hours for adolescents aged 14–17 [6]. These were the first evidence-based guidelines to integrate PA, SB, and sleep, providing a comprehensive framework for child and adolescent health [7,8]. A meta-analysis (MA) by Tapia-Serrano et al. [9], including 387,437 participants across 23 countries, reported a global adherence of only 2.68% among adolescents, with South America showing 2.93%. Gender differences were not significant, although adherence was lower in girls (1.86%) than in boys (3.54%). Conversely, non-adherence to all three guidelines reached 28.59% globally and 31.72% in South America, with higher prevalence among girls (14.79%) compared to boys (10.16%).

Despite the growing body of evidence on 24-HMB in adolescents [10], educational interventions targeting this age group remain scarce, and often lack the depth and rigor needed to achieve lasting impact [11,12]. In a recent systematic review (SR) of school-based interventions, only two randomized control trials conducted in adolescents were found [13]. Moreover, the complexity of 24-HMB which involves multiple interrelated behaviors, presents unique challenges for the design and implementation of effective interventions [14].

Self-Determination Theory (SDT) [15] offers a promising framework to address these challenges. By acknowledging the importance of the basic psychological needs of autonomy, competence, and relatedness, SDT provides a solid foundation for developing interventions that promote intrinsic motivation and sustained behavioral change [16]. Interventions grounded in SDT have been shown to be more effective because they move beyond external regulation, fostering self-endorsed forms of motivation that are more likely to persist over time [17,18]. Recent SR and MA highlights that school-based programs informed by theoretical frameworks, particularly SDT, are associated with greater improvements in PA, reduced SB, and enhanced psychological well-being among adolescents [19,20]. The rationale for proposing this intervention under SDT is that adolescents are more likely to engage in and maintain positive health behaviors when they perceive a sense of choice (autonomy) [21,22], feel capable of achieving goals (competence) [23,24], and experience meaningful

connections with others (relatedness) [25,26]. Thus, SDT not only guides the design of strategies that target these psychological needs but also provides an evidence-based pathway to strengthen adherence to 24-HMB in school settings.

Previous school-based interventions addressing 24-HMB in adolescents have shown promising, yet heterogeneous results. While multicomponent approaches that include family participation have demonstrated effectiveness in improving PA, ST, and sleep, evidence on the overall efficacy and sustainability of these interventions remains limited. For example, programs such as Kids'n Fitness [27] and ACTIVITAL [28], which integrated PA, SB, and dietary components with parental involvement, reported reductions in Body Mass Index, increases in PA, and decreases in ST. Similarly, interventions incorporating structured PA sessions within the school day or targeting sleep education have produced positive effects on health-related quality of life (HRQoL) [29], sleep duration [30], and psychosocial outcomes [29]. However, SR indicates that the effectiveness of these programs is variable, often constrained by short durations, limited sample sizes, and the lack of long-term follow-up [31]. Moreover, most studies tend to focus on isolated behaviors rather than adopting a fully integrated 24-HMB framework [32–36]. Recent evidence also highlights consistent associations between meeting multiple 24-HMB recommendations and better HRQoL in adolescents, underscoring the importance of comprehensive approaches [37,38]. Collectively, this emerging body of research supports the potential of school-based, theory-driven, multicomponent interventions, but also reveals important gaps regarding their integrated application and sustainability.

Within this context, the present study aimed to evaluate the effectiveness of a school-based intervention grounded in SDT on 24-HMB, adherence to 24-HMB recommendations, SDT-related constructs linked to 24-HMB, and HRQoL in secondary school students. It was hypothesized that the school-based intervention would have effects on 24-HMB, adherence to 24-HMB recommendations, SDT constructs related to the behaviors, and HRQoL.

## 2. Materials and methods

### 2.1. Design and participants

This study followed a quasi-experimental design following the TREND statement [39]. It was conducted over 12 weeks of intervention plus 12 weeks of follow-up, in two schools located in Bello, Colombia. Both schools had comparable academic schedules, instructional hours, and physical infrastructure, and were situated in neighborhoods with similar sociodemographic characteristics. Although both schools were located in the same city, separated by approximately 3.6 km, participants did not have direct contact between their daily activities, or shared facilities, teachers, or extracurricular programs. These conditions reduced the likelihood of information spillover between groups. The sample size was calculated based on the objective of increasing adherence to 24-HMB recommendations. Using the Epidat 3.1 software, a sample size of 23 students per group was estimated, assuming post-intervention adherence to 2 or 3 24-HMB recommendations based on proportions reported by Sevil-Serrano et al. [40], 65.6% in the experimental group (EG) and 17.6% in the control group (CG). Therefore, the minimum difference to be detected in compliance was 48% in favor of the experimental group. A confidence level of 95% and a statistical power of 90% were used in the calculation.

A convenience sample of 9th-grade students aged 14–17 years from each school was selected; one school constituted the EG (n = 39), and the other the CG (n = 43). Participants were required to be apparently healthy, and parental consent was obtained through signed informed consent forms. The ethics committee of the University Institute of Physical Education and Sport approved the study protocol (Approval ID: ACEI 46/2023). The study protocol was registered on the Open Science Framework with the following identifier: https://doi.org/10.17605/OSF.IO/CZMXA

### 2.2. Materials and instruments

Sociodemographic variables (age and sex) were collected through a printed, self-administered questionnaire. Height was measured in centimeters using a SECA 206 tape measure (SECA, Hamburg) mounted 2 meters high on a smooth wall, and weight was measured using a SECA 813 digital scale (SECA, Hamburg), with the result recorded in kilograms. Both measurements were taken twice, and the average value was used for analysis.

To assess 24-HMB, self-report questionnaires were administered. MVPA and sleep were measured using the *24-Hour Movement Behavior Questionnaire for Youth* [41], which assessed time spent in MVPA and sleep. Based on these data, adherence to the recommendations of at least 60 minutes of MVPA per day [42] and 8–10 hours of sleep [43] were analyzed. This questionnaire showed moderate reliability for weekdays, weekends, and total time (ICC 0.47 to 0.67). This instrument has been cross-culturally adapted in Colombia, demonstrating semantic and structural equivalence with the original English version, no validity across domains, and moderate reliability for weekdays, weekends, and total time [44].

ST was assessed using the *SAYCARE questionnaire* [45], which evaluates sedentary screen-based activities and determines adherence to the recommendation of two hours or less of screen time per day [42]. This questionnaire presents moderate test-retest reliability in children (rho ≥ 0.45 and k ≥ 0.40) and adolescents (rho ≥ 0.50) for self-report and showed low concurrent validity.

HRQoL was measured using the *KIDSCREEN-10 questionnaire* [46], which is validated for assessing HRQoL in adolescents. This questionnaire has a reliability of $\alpha = 0.78$ in the KIDSCREEN-10 version for children and adolescents.

For SDT constructs, adapted instruments were used to assess perceived competence, autonomy, and relatedness in relation to PA [47]. The overall hierarchical omega coefficient ($\omega$) for the NPB was 0.97, and the subscale omega coefficients ($\omega$) were 0.91, 0.91, and 0.94 for the specific factors of competence, autonomy, and relatedness, respectively; the *Motivation to Limit Screen-Time Questionnaire* was used to evaluate motivation to reduce ST [48]. with ICC values of 0.67 for amotivation and 0.70 and 0.82 for controlled and autonomous motivation, respectively; and the *Intrinsic Motivation Inventory* was used to assess perceived competence and the value of sleep [49].

## 2.3. Intervention

During the pre- and post-intervention assessments, the following were evaluated: 24-HMB, adherence to 24-HMB recommendations, SDT constructs related to 24-HMB, and HRQoL. A follow-up assessment was conducted 12 weeks after the intervention concluded for both groups, in which 24-HMB and adherence to the corresponding recommendations were measured. The study was carried out in the classroom and sports facilities of the participant schools.

The intervention was described according to TIDieR parameters [50] and was implemented by the physical education (PE) teacher between February and April 2024 and was carried out in the EG through the application and monitoring of four components: a) Educational component: two hours per week of theoretical-practical classes addressing the components of 24-HMB, framed within SDT constructs and emphasizing the benefits of PA, the implications of SB, sleep hygiene, and the 24-hour cycle and its importance for health; b) Family involvement: distribution of infographics to parents aimed at raising awareness about the practice and adherence to 24-HMB in their children; c) Recess-time intervention: weekly 30-minute sessions including playful activities, instructions, and tips to reduce SB; And d) Leisure-time activities: goal setting and planning around the development of healthy and physically active extracurricular habits. To ensure treatment fidelity, the PE teacher responsible for delivering the intervention was trained through a structured treatment manual that detailed the objectives, strategies, and examples of activities to be implemented during the PE sessions. In addition, the PE teacher registered his adherence to the prescribed activities after each session. The intervention was implemented in its entirety with the experimental group, and 100% of the planned activities were carried out as scheduled. After completing all assessments, the intervention will also be provided to the CG to ensure equitable access to its potential benefits. The same content and implementation procedures as described in the manual will be applied, thus respecting ethical standards and ensuring that all participants can benefit from the strategies designed to promote healthy movement habits. A more detailed description of the intervention components is provided in Table 1 and practical application of SDT constructs in Table 2.

The CG participated in regular PE classes (two hours per week for twelve weeks) following the school's standard curriculum and the didactic plan developed by the teacher. Topics covered included healthy habits, safety measures in PA, body hygiene and care, awareness of non-communicable chronic diseases, and team sports practice.

**Table 1. Discriminated distribution of interventions by components.**

| Component | Week | Topic | Time | Provider/location | Materials/procedures |
|---|---|---|---|---|---|
| **Educational component** | 1 | WHO recommendations on PA in adolescents. | 120 min | PE teacher/ Classroom and school court. | Projector, writing materials (including pens, pencils, and notebooks), and paper sheets. The activity was conducted through group work. |
| | 2 | Sedentary behavior and its health consequences. | 120 min | PE teacher/ Classroom. | Chairs, tables, mobile phones, and social media platforms. The activities were carried out both in groups and individually. |
| | 3 | Sleep and its characteristics. | 120 min | PE teacher. Classroom. | Mats, clocks, speakers, sleep diaries, eye masks, and pillows. The activities were carried out individually and in pairs. |
| | 4 | The 24-HMB and its importance for health. | 120 min | PE teacher. Classroom. | Visual aids (e.g., posters), instructional worksheets, whiteboard, dry-erase markers, and highlighters. The activities were carried out both in groups and individually. |
| | 5 | The different intensities of PA. | 120 min | PE teacher. Classroom and school court. | Projector, writing materials (including pens, pencils, and notebooks), and paper sheets. The activity was conducted through group work. |
| | 6 | Sedentary behavior and technological devices. | 120 min | PE teacher. Classroom. | Chairs, tables, mobile phones, and social media platforms. The activities were carried out both in groups and individually. |
| | 7 | Healthy sleep habits. | 120 min | PE teacher. Classroom. | Mats, clocks, speakers, sleep diaries, eye masks, and pillows. The activities were carried out individually and in pairs. |
| | 8 | The 24-HMB and its characteristics in daily life. | 120 min | PE teacher. Classroom. | Visual aids (e.g., posters), instructional worksheets, whiteboard, dry-erase markers, and highlighters. The activities were carried out both in groups and individually. |
| | 9 | The relationship between PA with space and time. | 120 min | PE teacher. Classroom and school court. | Projector, writing materials (including pens, pencils, and notebooks), and paper sheets. The activity was conducted through group work. |
| | 10 | The importance of limiting sitting time. | 120 min | PE teacher. Classroom. | Chairs, tables, mobile phones, and social media platforms. The activities were carried out both in groups and individually. |
| | 11 | Rest and sleep habits. | 120 min | PE teacher. Classroom. | Mats, clocks, speakers, sleep diaries, eye masks, and pillows. The activities were carried out individually and in pairs. |
| | 12 | Our body and its relationship with the 24-HMB in daily life. | 120 min | PE teacher. Classroom. | Visual aids (e.g., posters), instructional worksheets, whiteboard, dry-erase markers, and highlighters. The activities were carried out both in groups and individually. |
| **Family Approach** | 1, 4, 8, 12 | Awareness of the practice and adherence of the 24-HMB components in their children. | 10 min | PE teacher and parents/ Digital media and Home | Infographics and posters. Social network (e.g., Whatsapp) |
| **School recess** | 1–12 | PA promotion and reduction of sedentary behavior. | 30 min | PE teacher. School playground | Teaching and sports material (e.g., hoops, ropes, mats, cones). Directed games and recreational activities |
| **Leisure activities** | 1–12 | Extracurricular healthy habits and achievement of extra-class sports objectives. | Considering the amount of time students dedicate to activities | Student, coach, parents. park, soccer field, athletic track, neighborhood | Elements used autonomously for the practice of sports and physical activity. Purposes and goals established in the educational component regarding |

**Table 2. Behavior change techniques and strategies based on SDT constructs implemented in the school-based intervention program.**

| Construct | Mode of Implementation and Example | Type of Strategy (Need-supportive/ Controlling) |
|---|---|---|
| Autonomy | Throughout the intervention, students examined their own levels of physical activity, screen time, and sleep duration in relation to health recommendations. Discussions were held to capture students' individual perceptions. For instance: "Do you think you spend too much time on screens? Why?" | **Need-supportive (Autonomy)** – Promotes reflection, choice, and volition. |
| | Language used during the intervention avoided coercion. For example, facilitators used "you could" instead of "you should." Participation was not compulsory. For instance: "Would you like to join a sports event this weekend? We will have fun together." | **Need-supportive (Autonomy)** – Non-controlling language encourages volitional engagement. |
| | Students' interests and preferences were considered in the intervention design. For example: "Which activities would you like to engage in during school recess?" | **Need-supportive (Autonomy)** Acknowledges personal interests. |
| | A series of health challenges (e.g., commuting to school actively each day) were co-designed. Alternatives were varied, allowing each student to choose according to their individual interests and preferences. | **Need-supportive (Autonomy & Competence)** – Involves choice and personal adaptation. |
| | Sessions were conducted through games that foster interactions and creativity. For example, students identified new postures or motor challenges for active lessons. Additionally, facilitators provided information on opportunities and facilities for physical activity (e.g., "This park offers afternoon play sessions."). | **Need-supportive (Autonomy & Relatedness)** – Encourages creative participation and connectedness. |
| Competence | Students identified barriers to health-related behaviors (e.g., "What prevents you from being physically active?"). Then, they proposed potential solutions to increase activity levels. | **Need-supportive (Competence)** – Fosters problem-solving and mastery. |
| | Main expectations and challenges were identified to adapt the intervention design. For example, most students reported they wanted enjoyable activities, regardless of their intensity. | **Need-supportive (Competence)** – Adjusts difficulty to maintain engagement. |
| | Based on baseline assessments, students were encouraged to select short-term, realistic goals. Teachers and researchers supported them in adjusting these goals to their personal circumstances. For instance, students selected health challenges suited to their context. | **Need-supportive (Competence)** – Promotes mastery through attainable goals. |
| | Private, process-focused feedback was provided during the intervention. For example, during group discussions, teachers offered non-evaluative feedback based on student responses, aiming to enhance awareness of healthy behaviors. | **Need-supportive (Competence)** – Encourages learning and effort, not outcomes. |
| Relatedness | Research team members present during the intervention demonstrated empathy, understanding, and kindness. Students were allowed to privately express concerns or fears. | **Need-supportive (Relatedness)** – Builds trust and connection. |
| | At the start of each session, students verbalized their progress and health goals achieved. For example, they identified successful behaviors in front of peers. | **Need-supportive (Relatedness)** – Encourages mutual support and recognition. |
| | Students were reminded that success was defined by self-improvement rather than by comparison with others. For example, increasing one's physical activity time compared with the previous week was considered a success. Social comparisons were strictly avoided. | **Need-supportive (Competence & Relatedness)** – Reduces pressure and fosters internal motivation. |
| | Facilitators inquired about students' opinions on the program, their feelings, academic experiences, social context, and leisure interests. For instance, each session began with questions such as "How are you feeling today?" | **Need-supportive (Relatedness)** – Demonstrates genuine interest. |
| | Attention was given to students' reactions to videos and discussions. For example, after watching videos, debates among peers were encouraged so that students could generate answers independently. | **Need-supportive (Autonomy & Relatedness)** – Promotes expression and social connection. |
| | Health challenges could be performed with family members or friends to foster social support. For example, students could go cycling with their family during weekends. | **Need-supportive (Relatedness)** – Builds social support network. |

To minimize potential biases, several control strategies were implemented. To address selection bias due to dropout or absenteeism, students received follow-up phone calls to encourage continued participation, particularly in the EG. To prevent adaptation bias, the EG and CG were kept separate. Instrument-related biases were addressed through standardized measurement techniques, the use of validated and calibrated tools, questionnaire timing adjustments, breaks to reduce respondent fatigue, randomization of response categories, and the use of response scales to mitigate acquiescence and social desirability bias.

## 2.4. Data analysis

Parametric and non-parametric tests were used to compare baseline characteristics between groups. For age, weight, and height, data normality was assessed using the Shapiro–Wilk test (p > 0.05) and homogeneity of variances with Levene's test (p ≥ 0.05). Height was analyzed with parametric tests (Student's t-test) and summarized using means and standard deviations (SD). For age and weight, non-parametric tests (Mann–Whitney U) were applied, with medians and interquartile ranges (25th–75th percentiles) reported. Differences in the categorical variable sex were examined using the chi-square test, and group distributions were expressed as percentages. Mixed-effects models were applied to analyze the effect of the intervention on 24-HMB, SDT constructs, and HRQoL, adjusting for sex, age, weight, and height. Relative risk (RR) measures were calculated to assess the effect of the intervention on adherence to 24-HMB recommendations. Data analysis was conducted following the intention to treat. A $p$-value < .05 was considered statistically significant, with a 95% confidence level. Effect sizes were reported using the conditional R-squared ($R^2_c$) for mixed-effects models, which quantifies the variance explained by fixed and random effects; and the RR with 95% confidence intervals (CI) for binary outcomes, which provide a direct and interpretable measure of association. No correction methods for multiple comparisons were applied in the present analysis. All data analyses were performed using the open-source statistical software Jamovi (Version 2.6.25) [51].

## 3. Results

A total of 82 students participated in the study, distributed between two groups. The participant selection process is detailed in Fig 1. The demographic and anthropometric characteristics of the participants in each group are presented in Table 3. No statistically significant differences were found between the groups in terms of age, height, or weight. Likewise, no significant difference was observed in sex distribution. The average adherence to the intervention was 85.7% (SD = 23.3%) in the EG and 87.0% (SD = 11.1%) in the CG. No adverse events were reported during the activities conducted as part of the study.

### 3.1. Mixed models and fixed and random effects for 24-HMB

Table 4 presents the results of the mixed model analysis used to examine differences in 24-HMB (Sleep, MVPA and ST) between groups across the three assessment points (pre-test, post-test, and follow-up). For weekday sleep, no significant group-by-time interaction was found ($F(1.50, 156) = 2$, $p = 0.22$). However, for weekend sleep, a significant interaction between group and time was observed ($F(6.09, 156) = 2$, $p = .003$), suggesting that changes in weekend sleep patterns differed between the groups over time. Similarly, a significant interaction was observed for total weekly sleep (weekdays + weekends) ($F(3.88, 156) = 2$, $p = .02$), indicating differential change patterns between groups over time. Overall, sleep pattern behavior was inconsistent in CG, increasing in post-test and decreasing in the follow-up, while in EG, the intervention caused a progressive reduction in weekend and total sleep time.

For MVPA (weekly average), intragroup differences were found in the CG across time ($F(3.70, 156) = 2$, $p = .02$), with a decrease of 0.39 h/day from post-test to follow-up ($t(156) = 2.69$, $p < .01$). However, no significant group-by-time interaction was observed.

Regarding ST on weekdays, significant intragroup differences were found in the EG ($F(7.38, 156) = 2$, $p < .01$), indicating a notable reduction in screen use. The group-by-time interaction was also significant ($F(4.68, 156) = 2$, $p = .01$), suggesting that changes over time differed between groups. For weekend ST, significant reductions were observed in the EG ($F(11.29, 156) = 2$, $p < .01$), with a decrease of 2.12 h/day ($t(156) = 2.66$, $p < .01$) from pre- to post-test, and 1.66 h/day ($t(156) = 4.74$, $p < .001$) from post-test to follow-up. A significant group-by-time interaction was also found ($F(5.51, 156) = 2$, $p = .005$), with a 1.88 h/day reduction in the EG at follow-up, indicating again that ST changes differed between groups. For total weekly ST, significant reductions were observed in the EG of 1.73 h/day ($t(156) = 2.59$, $p = .01$) and 1.49 h/day

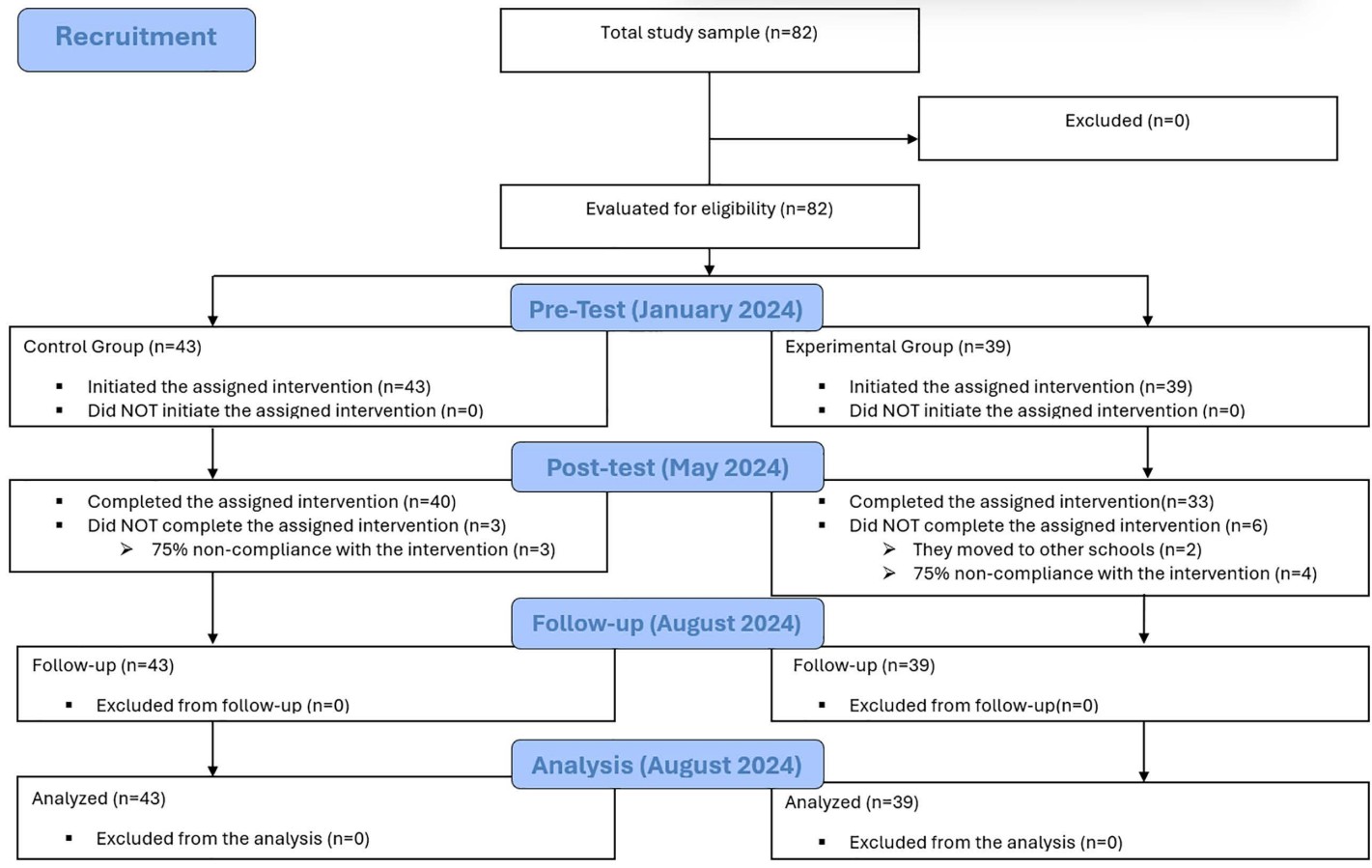

**Fig 1. Flowchart for the selection of study participants.**

**Table 3. Characteristics of participants.**

| Variables | Control Group (n = 43) | Experimental Group (n = 39) | p-value |
|---|---|---|---|
| Age (years)[a] | 14.8 (14.4-15.4) | 15.4 (14.7-15.9) | 0.13[c] |
| Height (cm)[b] | 161 (9.71) | 163.3 (7.78) | 0.24[d] |
| Weight (kg)[a] | 50.3 (44.5-55.4) | 52.3 (46.6-62.3) | 0.22[c] |
| Sex (% women) | 58.1 | 46.2 | 0.278[e] |

[a]Median (p25-p75); [b]Mean (SD); [c]Mann-Whitney U test; [d]Student T test; [e]Chi-square test; $p < 0.05$

($t(156) = 2.22$, $p = .02$) from pre- to post-*test* and from post-test to follow-up, respectively. A significant group-by-time interaction was found ($F(6.32, 156) = 2$, $p = .002$), with a 2.18 h/day reduction in total ST for the EG at follow-up ($t(190) = 2.75$, $p = .00$), indicating that *t*he EG significantly reduced its ST over time compared to the CG.

The marginal R-squared values ($R^2_m$) ranged from 0.08 to 0.22, indicating that the fixed effects (such as group or time of measurement) explained between 8% and 22% of the variance in the different behaviors (sleep, MVPA, and ST). The $R^2_c$ values ranged from 0.34 to 0.55, suggesting that the full model (including both fixed and random effects) explained

**Table 4. Mixed models for 24-hour movement behaviors, within- and between-group differences, interaction, and random effects.**

| Variables | Control Group (n=43) | | | | | Experimental Group (n=39) | | | | | Interaction Group x time | | Random effects | | | | |
|---|---|---|---|---|---|---|---|---|---|---|---|---|---|---|---|---|---|
| | Pre-test M (SE) | Post-test M (SE) | Follow-up M (SE) | F | p | Pre-test M (SE) | Post-test M (SE) | Follow-up M (SE) | F | p | F | p | $R^2_m$ | $R^2_c$ | ICC | LRT | p |
| **Sleep** | | | | | | | | | | | | | | | | | |
| Weekdays (h/d) | 7.63 (0.24) | 7.98 (0.24) | 7.96 (0.24) | 0.96 | 0.38 | 8.00 (0.25) | 8.21 (0.25) | 7.66 (0.25) | 1.79 | 0.16 | 1.5 | 0.22 | 0.08 | 0.37 | 0.31 | 20.7 | <.001 |
| Weekend (h/d) | 9.36[a] (0.30) | 10.22[b] (0.30) | 9.49[c] (0.30) | 4.01 | **0.02** | 10.93[a] (0.32) | 10.16[b] (0.32) | 10.04[d] (0.32) | 4.03 | **0.02** | 6.09 | **0.003** | 0.11 | 0.5 | 0.44 | 40.5 | <.001 |
| Total (h/d) | 8.50[a] (0.22) | 9.10[b] (0.22) | 8.73 (0.22) | 3.02 | 0.05 | 9.46[a] (0.23) | 9.18 (0.23) | 8.85[d] (0.23) | 2.84 | 0.06 | 3.88 | **0.02** | 0.1 | 0.46 | 0.4 | 34.2 | <.001 |
| **Physical activity** | | | | | | | | | | | | | | | | | |
| Weekdays (h/d) | 0.99 (0.15) | 1.03 (0.15) | 0.67 cd (0.15) | 2.57 | 0.07 | 0.92 (0.15) | 0.99 (0.15) | 0.95 (0.15) | 0.06 | 0.94 | 1.2 | 0.3 | 0.14 | 0.43 | 0.33 | 23.2 | <.001 |
| Weekend (h/d) | 0.89 (0.18) | 1.15 (0.18) | 0.73[ac] (0.18) | 2.4 | 0.09 | 1.04 (0.19) | 1.15 (0.19) | 1.26[a] (0.19) | 0.59 | 0.55 | 1.88 | 0.15 | 0.18 | 0.54 | 0.43 | 39.4 | <.001 |
| Total (h/d) | 0.94 (0.13) | 1.09 (0.13) | 0.70[ac] (0.13) | 3.7 | **0.02** | 0.98 (0.14) | 1.07 (0.14) | 1.10[a] (0.14) | 0.36 | 0.69 | 2.42 | 0.09 | 0.22 | 0.55 | 0.43 | 39.1 | <.001 |
| **Screen time** | | | | | | | | | | | | | | | | | |
| Weekdays (h/d) | 6.90 (0.57) | 7.24 (0.57) | 7.17[a] (0.57) | 0.14 | 0.86 | 7.34 (0.59) | 6.00 (0.59) | 4.68[ad] (0.59) | 7.38 | **<.001** | 4.68 | **0.01** | 0.1 | 0.39 | 0.32 | 21.4 | <.001 |
| Weekend (h/d) | 9.46 (0.63) | 9.91 (0.63) | 9.24[a] (0.63) | 0.4 | 0.66 | 11.14 (0.65) | 9.02[b] (0.65) | 7.36[acd] (0.65) | 11.29 | **<.001** | 5.51 | **0.005** | 0.11 | 0.34 | 0.26 | 14.6 | <.001 |
| Total (h/d) | 8.18 (0.54) | 8.57 (0.54) | 8.20[a] (0.54) | 0.23 | 0.78 | 9.24 (0.56) | 7.51[b] (0.56) | 6.02[acd] (0.56) | 11.65 | **<.001** | 6.32 | **0.002** | 0.12 | 0.38 | 0.3 | 19.1 | <.001 |

(h/d), hours per day; SE, standard error; [a] between-group differences; [b] pre-post differences; [c] post-follow-up differences; [d] pre-follow-up differences; models adjusted for sex, age, weight, and height; $R^2_m$, marginal R squared (fixed effects); $R^2_c$, conditional R squared (fixed and random effects); ICC, intraclass correlation coefficient; LRT, likelihood ratio test; statistical significance $p < .05$.

between 34% and 55% of the variance in the outcomes. The consistently higher $R^2_c$ compared to $R^2_m$ across all variables indicates that both fixed and random effects substantially contributed to the explained variance in the dependent variables.

The intraclass correlation coefficients (ICC) ranged from 0.26 (95% CI 1.71–6.99) to 0.44 (95% CI 0.96–2.44), indicating that between 26% and 44% of the total variance in 24-HMB was attributable to between-subject differences. High ICC values (>0.40) were observed for MVPA, weekend sleep, and total sleep, while moderate values (0.25–0.40) were found for week-day sleep, weekly MVPA, and ST. The likelihood ratio test showed that the random effect for subjects was significant in all models (p < .001), confirming sufficient inter-subject variability that justified the inclusion of random effects in the models.

### 3.2. Meeting with 24-HMB recommendations

Table 5 displays the intra- and intergroup differences and the RR of meeting the 24-HMB recommendations. Regarding sleep, the EG showed a significant increase in weekday sleep compliance post-intervention (56.4%) compared to base-line (38.5%) ($\chi^2$(1, N = 82) = 5.73, p = .01), suggesting a positive effect of the intervention. Additionally, post-intervention, the EG was 1.8 times more likely to meet the sleep recommendation (8–10 h/day) compared to the CG (RR = 1.85, 95% CI: 1.05–3.25). For MVPA, at follow-up, the EG demonstrated a significantly higher compliance rate during weekdays (48.7%) than the CG (25.6%) ($\chi^2$(1, N = 82) = 4.72, p = .03). However, the intervention did not have effect on the meeting of MVPA recommendation. No significant differences were found between groups in ST compliance at any assessment point. Likewise, there were no significant differences observed regarding compliance with one, two, or all three 24-HMB recommendations.

### 3.3. Mixed models for SDT constructs related to 24-HMB and HRQoL

Table 6 presents the SDT variables in relation to 24-HMB. In the EG, the competence construct in PA showed a signifi-cant increase ($t$(78) = −3.25, p = .002). The interaction was also significant ($F$(4.78, 78) = 1, p = .03), suggesting that the EG improved compared to the CG. The $R^2_c$ explained 84% of the variance in competence by including both fixed and random effects. For the relatedness construct, the EG showed a significant post-intervention increase ($t$(78) = −3.42, p < .001). Regarding motivation to limit ST, the CG showed a significant increase ($t$(78) = −2.99, p = .004) with a significant group × time interaction ($F$(7.56, 78) = 1, p = .007), indicating important differences in group trajectories. The $R^2_c$ of 48% suggests a moderate explanation of motivation variance when including fixed and random effects. No significant differ-ences were found between groups for SDT constructs related to sleep and HRQoL at any assessment time.

## 4. Discussion

This study evaluated the effectiveness of a school-based intervention grounded in SDT on 24-HMB, compliance with 24-HMB recommendations, SDT constructs related to 24-HMB, and HRQoL in secondary school students. The interven-tion partially improved 24-HMB, with reductions in ST and increased compliance with weekday sleep recommendations, but showed no significant effects on MVPA, SDT constructs of competence and relatedness, or HRQoL.

The intervention had partial effects in improving weekend and total sleep behaviors, as indicated by significant group × time interactions, with progressive reductions observed across pre-, post-intervention, and follow-up assessments in the EG. These findings align with Sevil-Serrano et al. [40], who reported similar results for weekend sleep and overall sleep behaviors.

The intervention was not effective for MVPA, consistent with findings by Tapia-Serrano et al. [52]. This may be due to the necessity of longer interventions (at least six months) and extracurricular practical activities such as sports tour-naments and educational outings to improve MVPA [53,54]. In contrast, Sevil-Serrano et al. [40] found significant MVPA improvements by integrating other knowledge areas and involving all educational community agents.

The intervention effectively reduced ST, as shown by group × time interaction analyses. After 12 weeks, the EG reduced ST by 1.34 h/day (weekdays), 2.12 h/day (weekend), and 1.73 h/day (total week), with improvements maintained at

**Table 5. Descriptive statistics of 24-hour movement behavior adherence, within- and between-group differences, and relative risk.**

| Variables | Pre-test | | | | Post-test | | | | | Follow-up | | | | |
|---|---|---|---|---|---|---|---|---|---|---|---|---|---|---|
| | Control Group n (%) | Experimental Group n (%) | $X^2$ | p | Control Group n (%) | Experimental Group n (%) | $X^2$ | p | RR (CI 95%) | Control Group n (%) | Experimental Group n (%) | $X^2$ | p | RR (CI 95%) |
| **Meeting sleep (8–10 h/d)** | | | | | | | | | | | | | | |
| Weekdays | 16 (37.2) | 15 (38.5) | 0.01 | 0.90 | 13 (30.2) | 22 (56.4)[a] | 5.73 | **0.01** | **1.85 (1.05-3.25)** | 18 (41.9) | 18 (46.2) | 0.15 | 0.69 | 1.09 (0.64-1.86) |
| Weekend | 19 (44.2) | 17 (43.6) | 0.00 | 0.95 | 21 (48.8) | 19 (48.7) | 1.16 | 0.99 | 1.04 (0.64-1.69) | 27 (62.8)[b] | 21 (53.8) | 0.67 | 0.41 | 0.82 (0.55-1.22) |
| Total | 21 (48.8) | 24 (61.5) | 1.33 | 0.24 | 20 (46.5) | 23 (59.0) | 1.27 | 0.25 | 1.31 (0.84-2.03) | 27 (62.8) | 27 (69.2) | 0.37 | 0.53 | 1.11 (0.78-1.57) |
| **Meeting MVPA (≥1 h/d)** | | | | | | | | | | | | | | |
| Weekdays | 16 (37.2) | 16 (41.0) | 0.12 | 0.72 | 21 (48.8) | 17 (43.6)[a] | 0.22 | 0.63 | 0.72 (0.43-1.20) | 11 (25.6) | 19 (48.7)[b] | 4.72 | **0.03** | 1.75 (0.93-3.31) |
| Weekend | 14 (32.6) | 16 (41.0) | 0.63 | 0.42 | 18 (41.9)[a] | 20 (51.3)[a] | 0.73 | 0.39 | 1.16 (0.72-1.89) | 14 (32.6) | 18 (46.2)[b] | 1.59 | 0.20 | 1.49 (0.85-2.60) |
| Total | 17 (39.5) | 16 (41.0) | 0.01 | 0.89 | 17 (39.5)[a] | 18 (46.2)[a] | 0.36 | 0.54 | 1.04 (0.60-1.79) | 12 (27.9) | 19 (48.7)[b] | 3.77 | 0.05 | 1.81 (0.99-3.29) |
| **Meeting screen time (≤2 h/d)** | | | | | | | | | | | | | | |
| Weekdays | 4 (9.3) | 5 (12.8) | 0.25 | 0.61 | 8 (18.6) | 4 (10.3) | 1.14 | 0.28 | 0.59 (0.18-1.94) | 5 (11.6) | 8 (20.5) | 1.21 | 0.27 | 1.81 (0.58-5.62) |
| Weekend | 1 (2.3) | 2 (5.1) | 0.45 | 0.50 | 1 (2.3) | 1 (2.6) | 0.00 | 0.94 | 0.52 (0.01-15.9) | 1 (2.3) | 4 (10.3) | 2.25 | 0.13 | 4.72 (0.53-42.08) |
| Total | 0 (0.0) | 1 (2.6) | 1.12 | 0.29 | 0 (0.0) | 1 (2.6) | 1.12 | 0.29 | N/A | 2 (4.7) | 5 (12.8) | 1.75 | 0.18 | 2.96 (0.58-14.91) |
| **24-HMB guidelines** | | | | | | | | | | | | | | |
| Meeting no recommendations | 14 (32.6) | 9 (23.1) | 1.12 | 0.57 | 13 (30.2) | 11 (28.2) | 3.98 | 0.26 | N/A | 11 (25.6) | 5 (12.8) | 5.48 | 0.14 | N/A |
| Meeting one recommendation | 20 (46.5) | 19 (48.7) | | | 23 (53.5) | 15 (38.5) | | | | 23 (53.5) | 20 (51.3) | | | |
| Meeting two recommendations | 9 (20.9) | 11 (28.2) | | | 7 (16.7) | 12 (30.8) | | | | 9 (20.9) | 11 (28.2) | | | |
| Meeting three recommendations | 0 (0.0) | 0 (0.0) | | | 0 (0.0) | 1 (2.6) | | | | 0 (0.0) | 3 (7.7) | | | |

MVPA, Moderate and vigorous physical activity; (h/d), hours per day; $X^2$, chi-square; RR, Relative risk (models adjusted for sex, age, weight, and height); [a]pre-post differences; [b]post-follow-up differences; N/A, Not applicable; statistical significance $p < .05$.

**Table 6. Mixed models for SDT constructs and HRQoL, within- and between-group differences, interaction and random effects.**

| Variables | Control Group | | | | | Experimental Group | | | | | Interacción Group x time | | Random effects | | | | |
|---|---|---|---|---|---|---|---|---|---|---|---|---|---|---|---|---|---|
| | Pre-test M (SE) | Post-test M (SE) | MD | SE | p | Pre-test M (SE) | Post-test M (SE) | MD | SE | p | F | p | $R^2_m$ | $R^2_c$ | ICC | LRT | p |
| **SDT variables** | | | | | | | | | | | | | | | | | |
| **Physical activity** | | | | | | | | | | | | | | | | | |
| Autonomy (1–7) | 3.93 (0.21) | 4.15 (0.22) | 0.21 | 0.21 | 0.30 | 4.16 (0.22) | 4.43 (0.22) | 0.26 | 0.21 | 0.22 | 0.02 | 0.87 | 0.10 | 0.56 | 0.51 | 23.6 | <.001 |
| Competence (1–7) | 4.35 (0.22) | 4.39 (0.22) | 0.03 | 0.15 | 0.82 | 3.95 (0.23) | 4.47[a] (0.23) | 0.51 | 0.15 | **0.002** | 4.79 | **0.03** | 0.28 | 0.84 | 0.77 | 70.8 | <.001 |
| Relatedness (1–6) | 3.72 (0.16) | 3.83 (0.16) | 0.11 | 0.12 | 0.38 | 3.69 (0.17) | 4.15[a] (0.17) | 0.45 | 0.13 | **<.001** | 3.45 | 0.06 | 0.22 | 0.76 | 0.69 | 50.8 | <.001 |
| **Screen time** | | | | | | | | | | | | | | | | | |
| Motivation (1–7) | 3.38 (0.13) | 3.81[a] (0.13) | 0.43 | 0.14 | **0.004** | 3.67 (0.14) | 3.53 (0.14) | 0.14 | 0.15 | 0.35 | 7.56 | **0.007** | 0.06 | 0.48 | 0.44 | 17.1 | <.001 |
| **Sleep** | | | | | | | | | | | | | | | | | |
| Competence (1–7) | 4.01 (0.19) | 4.30 (0.19) | 0.29 | 0.19 | 0.13 | 4.14 (0.19) | 4.18 (0.19) | 0.03 | 0.20 | 0.85 | 0.82 | 0.36 | 0.08 | 0.51 | 0.47 | 19.4 | <.001 |
| Value (1–7) | 5.69 (0.19) | 5.58 (0.19) | 0.11 | 0.19 | 0.56 | 5.40 (0.19) | 5.77 (0.19) | 0.37 | 0.20 | 0.06 | 3.07 | 0.08 | 0.06 | 0.52 | 0.49 | 21.2 | <.001 |
| **HRQoL** | | | | | | | | | | | | | | | | | |
| HRQoL index (1–100) | 62.5 (2.02) | 62.3 (2.02) | 0.20 | 1.67 | 0.90 | 66.1 (2.10) | 65.8 (2.10) | 0.31 | 1.73 | 0.85 | 0.00 | 0.96 | 0.25 | 0.74 | 0.65 | 43.4 | <.001 |

SE, standard error; MD, mean differences; [a] pre-post differences; models adjusted for sex, age, weight, and height; $R^2_m$, marginal R squared (fixed effects); $R^2_c$, conditional R squared (fixed and random effects); ICC, intraclass correlation coefficient; LRT, likelihood ratio test; statistical significance $p < .05$.

follow-up. These findings resemble patterns observed by Sevil-Serrano et al. [40] in a year-long intervention. The present study's effectiveness was achieved with a shorter exposure time compared to long-term interventions focusing on ST reduction [28,55]. The effectiveness may be attributable to intervention strategies targeting ST reduction, such as device use restrictions during school hours and awareness campaigns for students and parents. Evidence indicates that incremental decreases in screen exposure are associated with measurable improvements in physical, mental, and social health outcomes, including reduced risk of obesity, enhanced sleep quality, and improved psychosocial well-being [56]. From a public health perspective, such reductions represent a substantial behavioral shift, given the typically high baseline levels of ST in this population. Importantly, these findings suggest that even partial adherence to the 24-HMB ST recommendation can yield tangible health benefits [4], reinforcing the value of school-based interventions that aim not only for full compliance but also for gradual, realistic reductions in SB among adolescents.

The intervention increased compliance with weekday sleep recommendations by 17.9 percentage points between pre- and post-intervention ($p < .05$) in the EG, similar to the 15.2-point increase reported by Sevil-Serrano et al. [40]. School schedules likely facilitate stable sleep patterns during weekdays [57], and this may have been reinforced by parental control, as parents received information about strategies of how to improve the 24-HMB of participants. For instance, sleep-related strategies implemented with parents included infographics about awareness, knowledge, hygiene, and sleep health effects.

Consistent with continuous variable analyses, no effectiveness was found for MVPA, differing from Sevil-Serrano et al. [40], who documented increased MVPA compliance. Similarly, no significant differences were observed between groups in ST compliance post-intervention, consistent with Tapia-Serrano et al. [52]. Conversely, Sevil-Serrano et al. [40] reported significant ST compliance improvements after 12 months. Despite absolute ST reductions, no significant compliance changes (≤ 2 h/day) were detected, possibly due to variability in assessment tools ranging from TV-only questionnaires to broader digital device usage surveys.

Changes were found in SDT competence and relatedness constructs related to PA in the EG, consistent with Gonzalez-Cutre et al. [18], who reported favorable changes in all SDT constructs with a different intervention design (three 90-minute sessions per week over 30 weeks) involving parental participation. SR and MA [58,59] suggest small, statistically nonsignificant effect sizes for SDT-based PA interventions. Motivation related to ST did not explain ST reduction in the EG, aligning with limited evidence on SDT use in ST reduction [60,61].

No significant intervention effects were observed for HRQoL. To our knowledge, no previous interventions have examined HRQoL as an outcome within the framework of 24-HMB, which limits the possibility of direct comparisons. Nevertheless, prior studies have shown that changes in PA can positively influence HRQoL among adolescents, particularly in interventions lasting between 2 and 12 months [62]. In contrast, the relatively short duration of the present program, coupled with potential limitations in study design and the sensitivity of the measurement instruments employed, may have hindered the detection of meaningful effects. These considerations underscore the importance of conducting further research with longer intervention periods, more robust methodological designs, and culturally sensitive instruments to clarify the causal relationships between 24-HMB and HRQoL.

Few experimental studies have been documented in literature and conducted to identify the effect of interventions on 24-HMB. Only two studies have been identified in children and adolescents [40,52]. However, when comparing the findings with those reported in the present study, it is worth noting that caution should be exercised considering the gradient of influence of family dynamics, school routines, cultural norms, and age group on 24-HMB in different geographic and sociocultural contexts. Therefore, it is important to tailor the interventions not only to cultural contexts but also to school environments and developmental stages.

Findings of the present study could be informative for strengthening current policies aimed at school health and formulating new ones. For example, in Colombia, PE is a mandatory subject at all levels of basic education. Therefore, its implementation is feasible. Additionally, the findings can enhance the strategies included in the "Healthy educational

environment program" [63], since this program encourages parental involvement as an essential component of the educational environment.

### 4.1. Limitations and strengths

The study presents some limitations that must be acknowledged. First, the quasi-experimental design, potentially affecting internal validity. This convenience allocation may have introduced selection bias, as unmeasured contextual factors (school environment, teaching practices and contents), may partly explain the intervention effects. Second, the sample size was calculated considering information from a prior study and did not adjust for the clustering effect at the school level. This limitation may have affected the effective sample size and impacted the precision of the estimates. Third, the use of self-report questionnaires is related to information bias, which may result in over- or under-estimation of actual behaviors. Consequently, the precision of the observed effects may be limited. Future studies should incorporate objective measurements tools, such as accelerometers to strengthen validity. Fourth, taking into account that the study was conducted in an area of low socioeconomic position, findings present limited generalizability. And fifth, three separated mixed models were analyzed, without adjustments for multiple comparisons. This approach may increase the overall risk of Type I error. Future research with larger samples should consider adjustments when testing multiple outcomes simultaneously.

The strengths of the study include implementation in a real-world setting, family involvement, use of SDT as a theoretical framework, robust analytical methods, and a combination of educational, familial, and extracurricular strategies.

### 4.2. Suggestions for future research

Findings of the study allow to identify some aspects to be taken into account in future studies: 1) extending intervention duration to assess effectiveness on MVPA; 2) including and analyzing effects on other mental, physical, and social health markers; 3) investigating external school factors influencing intervention effectiveness on 24-HMB; and 4) exploring implementation processes to enhance external validity.

## 5. Conclusion

The school-based intervention partially improved 24-HMB in secondary school students, primarily by reducing ST and increasing compliance with weekday sleep recommendations. Positive effects were also observed in SDT competence and relatedness constructs associated with PA, although MVPA improvements were not significant. No significant post-intervention changes were found in HRQoL. Such incremental improvements are particularly relevant in school contexts, where realistic and sustainable strategies are essential for promoting healthier movement behaviors. Future research should build on these insights by examining longer-term interventions, testing scalable implementation models, and integrating multicomponent strategies that address the interconnected nature of sleep, PA, and ST. Strengthening the evidence base in this way will not only advance theoretical understanding but also guide the design of school-based programs that are both feasible and impactful in improving adolescent well-being.

## Supporting information

**S1 File. Database.**
(XLSX)

## Acknowledgments

We thank the selected schools and the Municipal Education Secretariat of Bello, Colombia, for providing access and facilitating the implementation of this project.

## Author contributions

**Conceptualization:** Jaminson Raul Ricardo-Sejin, Carlos Mario Arango-Paternina, Fredy Alonso Patino-Villada.

**Data curation:** Fredy Alonso Patino-Villada.

**Formal analysis:** Jaminson Raul Ricardo-Sejin, Carlos Mario Arango-Paternina, Fredy Alonso Patino-Villada.

**Investigation:** Jaminson Raul Ricardo-Sejin.

**Methodology:** Jaminson Raul Ricardo-Sejin, Carlos Mario Arango-Paternina, Fredy Alonso Patino-Villada.

**Project administration:** Jaminson Raul Ricardo-Sejin.

**Resources:** Jaminson Raul Ricardo-Sejin.

**Supervision:** Carlos Mario Arango-Paternina, Fredy Alonso Patino-Villada.

**Writing – original draft:** Jaminson Raul Ricardo-Sejin, Carlos Mario Arango-Paternina, Fredy Alonso Patino-Villada.

**Writing – review & editing:** Jaminson Raul Ricardo-Sejin, Carlos Mario Arango-Paternina, Fredy Alonso Patino-Villada.

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
