## [Decision Letter · Decision Letter 0]

19 Aug 2025

Dear Dr. Ricardo-Sejin,

Thank you for submitting your manuscript to PLOS ONE. After careful consideration, we feel that it has merit but does not fully meet PLOS ONE’s publication criteria as it currently stands. Therefore, we invite you to submit a revised version of the manuscript that addresses the points raised during the review process.

We look forward to receiving your revised manuscript.

Kind regards,

Henri Tilga, PhD

Academic Editor

PLOS ONE

3. In the online submission form, you indicated that [Our circumstances are not covered by the questions above and you need the journal’s help to make your data available.].

Additional Editor Comments (if provided):

Reviewers' comments:

Reviewer's Responses to Questions

**Comments to the Author**

1. Is the manuscript technically sound, and do the data support the conclusions?

Reviewer #1: Yes

Reviewer #2: Yes

Reviewer #3: Yes

2. Has the statistical analysis been performed appropriately and rigorously?

Reviewer #1: Yes

Reviewer #2: Yes

Reviewer #3: Yes

3. Have the authors made all data underlying the findings in their manuscript fully available?

Reviewer #1: Yes

Reviewer #2: No

Reviewer #3: No

4. Is the manuscript presented in an intelligible fashion and written in standard English?

Reviewer #1: Yes

Reviewer #2: Yes

Reviewer #3: Yes

Reviewer #1: I would like to thank for the opportunity to review this manuscript. Please see the following comments to consider to further increase the quality of this manuscript.

Strengths:

The integration of 24-HMB within an SDT framework is innovative, particularly in a Latin American adolescent population.

The inclusion of educational, family, recess-time, and leisure components addresses multiple ecological levels.

The 12-week post-intervention assessment allows examination of sustainability of effects.

Registration on OSF and adherence to TREND and TIDieR guidelines enhance reproducibility.

Use of mixed-effects models with random effects is appropriate for repeated-measures clustered data.

Suggestions for improvement:

The manuscript states that one school was the experimental group (EG) and another the control group (CG). This convenience allocation risks baseline differences and school-level confounding. Explicitly acknowledge this in the limitations and discuss how it may have affected internal validity.

Given the proximity and similarity of the schools, describe measures taken to avoid information spillover between groups.

The sample size calculation is based on proportions from a prior study but does not account for clustering at the school level, which likely reduces effective sample size. This should be noted as a limitation.

All behavioral measures (MVPA, sleep, screen time) were self-reported, which can over- or under-estimate true values. This should be more explicitly acknowledged in the limitations, with discussion of potential implications.

While you cite validation studies, specify whether these instruments were culturally adapted/validated for Colombian youth or Spanish-speaking populations in Latin America.

The lack of accelerometer-based data is a key limitation, particularly for detecting changes in physical activity.

While p-values are provided, effect sizes (e.g., Cohen’s d, partial η², or odds ratios with 95% CIs) should be reported for main outcomes to contextualize practical significance.

Given the large number of statistical tests, discuss whether adjustments (e.g., Bonferroni, FDR) were considered to control Type I error.

The discussion sometimes implies meaningful trends where p-values are >.05. These should be interpreted cautiously, avoiding overstatement.

The intervention is detailed in Table 1, but its theoretical integration with SDT could be made more explicit. For example:

How did each activity target autonomy, competence, or relatedness? Please see a recent manuscript by Ahmadi et al., (2023) in which a classification system for autonomy, competence and relatedness support techniques was developed. I think this manuscript would benefit from insights from Ahmadi et al., (2023) paper.

Ahmadi, A., Noetel, M., Parker, P., Ryan, R. M., Ntoumanis, N., Reeve, J., Beauchamp, M., Dicke, T., Yeung, A., Ahmadi, M., Bartholomew, K., Chiu, T. K. F., Curran, T., Erturan, G., Flunger, B., Frederick, C., Froiland, J. M., González-Cutre, D., Haerens, L., . . . Lonsdale, C. (2023). A classification system for teachers’ motivational behaviors recommended in self-determination theory interventions. Journal of Educational Psychology, 115(8), 1158–1176. https://doi.org/10.1037/edu0000783

Were teachers trained in autonomy-supportive behaviors, and if so, how was fidelity monitored?

Consider summarizing the four components more concisely in the main text and moving full weekly details to supplementary material.

The discussion could better link observed behavioral changes (e.g., in screen time and sleep) to the specific intervention strategies and SDT constructs.

While possible explanations are given (short duration, lack of extracurricular sports), consider also whether barriers in the school or community context were at play.

Deepen the contrast between your findings and similar interventions in different sociocultural contexts.

Add a brief paragraph on how findings could inform school health policies in Colombia or Latin America.

Reviewer #2: The purpose of this study is to evaluate the effectiveness of a school-based intervention grounded in Self-Determination Theory on: (1) 24-hour movement behaviors; (2) adherence to 24-hour movement behaviors recommendations; (3) health-related quality of life; and (4) Self-Determination Theory constructs associated with 24-hour movement behaviors.

The authors utilized a quasi-experimental study design with experimental and control groups.

Below are my suggestions and criticisms:

• In the abstract and introduction, a description and definition of Self-Determination Theory is needed.

• In the abstract, quantitative results should be included in the abstract.

• In the introduction, a description of the public health burden is needed. For example, the prevalence of adolescents who meet the physical activity, sedentary behavior, and sleep recommendations.

Methods:

• What minimum difference in outcomes is the sample size/power able to detect?

• Why couldn’t the authors utilize a randomized control trial?

• More of the details of the validity and the reliability of each of the 24-Hour Movement Behaviors.

• For the intervention, how was Self-Determination Theory used to develop the intervention?

Results:

• At the top of page 11, lines 185 to 188, could the authors describe the direction in the measure of association?

Discussion:

• The Discussion often repeats the findings described in the results section. It would be helpful if the authors could describe how the findings not only fit in with other studies but also explain any public health implications.

• Could the authors explain the limitations in the context of the current study. How did the quasi-experimental design affect internal validity, use of self-report questionnaires as being prone to self-report bias.

Reviewer #3: Effects of a school-based intervention on 24-hour movement behaviors in adolescents: A quasi-experimental study

I would like to sincerely thank the authors for the thorough and careful work carried out in addressing the reviewers’ comments. The manuscript entitled “Effects of a school-based intervention on 24-hour movement behaviors in adolescents: A quasi-experimental study” represents a valuable contribution to the scientific literature, particularly in the field of adolescent health promotion and educational interventions.

Abstract

The summary clearly presents the objectives, design, and main results, highlighting the effects on sleep and screen time, as well as some improvements in motivational constructs. However, the effects of the intervention are not shown. I suggest adding the main results of your study.

Furthermore, it would be advisable to highlight the practical value of these partial improvements and conclude with a more explicit projection towards future research and the applicability of these interventions in school contexts.

Introduction

The theoretical framework is well-founded and structured. It presents the current state of the art on the subject under study and justifies the need to address the problem.

However, there are a few points I would like to highlight in this section that would be useful for improving the understanding of this heading. First, the authors should improve the connection between paragraphs. There are too many jumps between ideas from one paragraph to another, and the explanations are very brief. It would be necessary to expand this section a little more.

In addition, it would be interesting to include a brief paragraph on compliance with these recommendations, when they were established, and what those guidelines are. Here, the study by Tremblay et al. (2016) should be mentioned. Next, in this same paragraph, the authors should include a paragraph on the prevalence of compliance with the 24-hour recommendations for each of these behaviours. This would further reinforce the need to address this problem. For this paragraph, they could rely on the citation from Tapia-Serrano et al. (2022). Here are both references:

Tremblay, M. S., Carson, V., Chaput, J. P., Connor Gorber, S., Dinh, T., Duggan, M., ... & Zehr, L. (2016). Canadian 24-hour movement guidelines for children and youth: an integration of physical activity, sedentary behaviour, and sleep. Applied physiology, nutrition, and metabolism, 41(6), S311-S327.

Tapia-Serrano, M. A., Sevil-Serrano, J., Sanchez-Miguel, P. A., Lopez-Gil, J. F., Tremblay, M. S., & Garcia-Hermoso, A. (2022). Prevalence of meeting 24-Hour Movement Guidelines from pre-school to adolescence: A systematic review and meta-analysis including 387,437 participants and 23 countries. Journal of sport and health science, 11(4), 427-437.

In addition, I have found that some important references are missing from the introduction, which I will detail below.

The reference to Rollo et al. (2020) is important to include in the first paragraph, where they point out the effects of the 24-hour recommendations on young people's health, as this review has a specific section to clarify these findings. I am attaching the reference:

Rollo, S., Antsygina, O., & Tremblay, M. S. (2020). The whole day matters: understanding 24-hour movement guideline adherence and relationships with health indicators across the lifespan. Journal of sport and health science, 9(6), 493-510.

It is important to mention the reference to Rodrigo-Sanqoquín in the section on the effect of school interventions based on the 24-HMB, as it is the only review that analyses the effect of these school interventions. I am attaching the reference:

Rodrigo-Sanjoaquín, J., Tapia-Serrano, M. Á., López-Gil, J. F., & Sevil-Serrano, J. (2025). Effects of school-based interventions on all 24-hour movement behaviours in young people: a systematic review and meta-analysis of randomised controlled trials. BMJ Open Sport & Exercise Medicine, 11(2).

The section on SDT is very brief. I do not fully understand the reasons why you decided to propose this intervention under the postulates of this theoretical framework. There are studies that suggest that interventions based on theoretical frameworks are more effective, but these reviews are not cited in your document. Please explain this postulate better and add the citations on which you base this claim.

Finally, the state of the art on the effect of 24-HMB behaviours is unclear. Just before presenting the objectives and hypotheses, the authors should add a paragraph showing previous interventions in adolescents and based on those results, add what led them to consider this study. In other words, why are they analysing the effect of 24-HMB on quality of life and not on other variables? I imagine there is previous evidence on which they based their decision. This would reinforce the idea behind their approach and justify their intervention; otherwise, these are very general and vague ideas.

Method

The authors state: ‘It was conducted over 24 weeks (12 weeks of intervention and 12 weeks of follow-up) in two schools located in Bello, Colombia.’ This sentence is incorrect. Their intervention lasts 12 weeks, which is the time during which they are intervening. The other 12 weeks are a follow-up period, but this does not count towards the duration of the intervention. Modify this and revise the rest of the document accordingly.

Review and improve the section on instruments. It is not correct to include the KIDSCREEN questionnaire and the motivation questionnaire in the same paragraph. Ideally, you should add a paragraph for each variable and instrument.

Expand the information in this section. You must add the factor loadings and Cronbach's alpha for each instrument to make it correct. Also add how the factors for each instrument are formed. I suggest you look at some sample articles for support and improve this section. It is very poor and unscientific.

Did you also fill out the questionnaires? What actions and strategies were implemented? Once the intervention was completed, were they offered the opportunity to carry out the intervention so that they could benefit from the programme? Please add this information to the intervention section.

The statistical analysis section is solid but could be improved in several ways. First, although it is indicated that a significance threshold of p < .05 was used, the effect sizes are not reported. This element is essential for assessing the practical magnitude of the findings and not just their statistical significance. Including measures such as Cohen's d, partial η², or the marginal and conditional R² values of the mixed linear models would provide a much more complete context for interpreting the results.

On the other hand, when analysing multiple variables related to sleep, physical activity, screen time, ADHD constructs and quality of life, the risk of type I error increases. However, it is not specified whether any correction method for multiple comparisons, such as Bonferroni, Holm or FDR, was applied. Making this aspect explicit would reinforce the robustness of the analysis.

It would also be advisable to justify in greater detail the use of parametric and non-parametric tests. The text states that both were used for baseline comparisons but does not clarify which variables were analysed with each test or under what criteria (such as normality of data or homogeneity of variances). Clarification in this regard would provide greater methodological transparency.

Please address these issues in this section. I suggest that you support and/or revise this section based on some previous intervention studies, as it is too brief and leaves many questions unanswered for readers.

Results

Looking at the results for the screen time variable, we can see that this variable has been overestimated. Although the questionnaire used to measure screen time is valid and reliable, its calculation of the resulting average time is excessive and unrealistic, which could affect the accuracy of the results.

For example, the authors report that the total screen time of the participants in the experimental group (which can also be extended to the control group) is 7.51 hours/day (Table 3). This seems excessive and unrealistic for children and adolescents, suggesting a possible overestimation by the participants. It is suggested that the maximum values be reviewed to identify possible discrepancies that could bias the results.

Discussion

The discussion contains some redundancies, especially in the explanation of the results related to moderate to vigorous physical activity and screen time, which could be summarised to improve the flow of the text. In some passages, the tone tends to overestimate the effectiveness of the intervention, so it would be more appropriate to qualify the language and refer to partial or limited effects, so that the results are aligned with the actual evidence.

The absence of effects on health-related quality of life is mentioned briefly, but it would be useful to discuss in greater depth whether this was due to limitations in the design, the measurement instrument used, or the short duration of the intervention. It would also be advisable to reflect further on the role of school and family contexts in the sustainability of the changes observed in 24-hour behaviours, as their influence can be decisive in maintaining long-term effects. Finally, the section could be enriched with a more practical analysis of the findings, highlighting that reductions of between 1.5 and 2 hours per day in screen time are clinically relevant for adolescent health, even if the recommendation of two hours or less per day is not fully achieved.

Finally, the section on limitations, strengths and suggestions for future studies should be restructured under the same heading.

Conclusion

The conclusion summarises the findings well, highlighting improvements in sleep, reduced screen time and positive effects on some motivational constructs. However, the interpretation is limited, as it does not clarify that the effects were partial and that there were no significant improvements in physical activity or quality of life. It would be advisable to emphasise the practical value of modest changes in sleep and screen use, as well as to conclude with a clearer projection towards future research and application in school contexts.

References.

Please review the references section as they appear to be in a different font.

Also review Figure 1, as it is of poor quality.

**Do you want your identity to be public for this peer review?** For information about this choice, including consent withdrawal, please see our Privacy Policy

Reviewer #1: No

Reviewer #2: No

Reviewer #3: No

---

## [Author Response · Author response to Decision Letter 1]

6 Oct 2025

We thank to the editor and reviewers for the suggestions made. We have addresed all the reviewers comments

---

## [Decision Letter · Decision Letter 1]

8 Oct 2025

Dear Dr. Ricardo-Sejin,

Thank you for submitting your manuscript to PLOS ONE. After careful consideration, we feel that it has merit but does not fully meet PLOS ONE’s publication criteria as it currently stands. Therefore, we invite you to submit a revised version of the manuscript that addresses the points raised during the review process.

We look forward to receiving your revised manuscript.

Kind regards,

Henri Tilga, PhD

Academic Editor

PLOS ONE

Journal Requirements:

Reviewers' comments:

Reviewer's Responses to Questions

**Comments to the Author**

Reviewer #1: All comments have been addressed

Reviewer #3: All comments have been addressed

2. Is the manuscript technically sound, and do the data support the conclusions?

Reviewer #1: Yes

Reviewer #3: Yes

3. Has the statistical analysis been performed appropriately and rigorously?

Reviewer #1: Yes

Reviewer #3: Yes

4. Have the authors made all data underlying the findings in their manuscript fully available?

Reviewer #1: Yes

Reviewer #3: Yes

5. Is the manuscript presented in an intelligible fashion and written in standard English?

Reviewer #1: Yes

Reviewer #3: Yes

Reviewer #1: Authors have done well job on revising their manuscript. Well job! Manuscript is ready for the publication.

Reviewer #3: Effects of a school-based intervention on 24-hour movement behaviors in adolescents: A quasi-experimental study

Firstly, I would like to thank the editor for the opportunity to review your manuscript entitled ‘Effects of a school-based intervention on 24-hour movement behaviors in adolescents: A quasi-experimental study’. I appreciate the opportunity to review your work, which addresses a topic of great relevance to public health and adolescent well-being. The study has the potential to make an important contribution in the field of school intervention and the promotion of healthy habits.

Below, I present my comments and suggestions, which I hope will contribute to improving the quality and clarity of the article.

Strengths of the manuscript

The study is well designed, with an appropriate methodological structure that allows for rigorous evaluation of the effects of the school intervention. The use of Self-Determination Theory (SDT) as a theoretical framework is appropriate, and the quasi-experimental approach with control and experimental groups provides a solid basis for the analyses presented. Furthermore, the topic of 24-hour movement behaviours (physical activity, sedentary behaviour, and sleep) is highly topical and relevant, given the growing interest in the integration of these three aspects in adolescent health.

Minor comments

Despite the strengths mentioned above, there are several aspects that could benefit from further revision to improve the clarity and depth of the analysis. Below are some suggestions and observations:

Clarification on support and control strategies in SDT

One of the most relevant points in this study is the use of SDT-based strategies. However, I would like to suggest clarifying whether the autonomy, competence, and relatedness strategies implemented in the intervention are primarily aimed at supporting these psychological needs or controlling them. According to SDT, these needs can be met in ways that either support or control individuals' motivation, and it is important to highlight this point in the article, as it could have implications for the interpretation of the results. This distinction is fundamental to understanding how the intervention impacts participants' motivation and behaviours.

Updating the tables

While I appreciate the authors' response to the reviewers' comments, I must point out that, for future occasions, it would be advisable to send a supplementary response to the reviewers' comments. The format in which the authors have responded to the comments makes it difficult to review the article, as the changes made to the manuscript are not clearly specified, nor are the exact sections that have been changed or added indicated. A more detailed and structured response would help reviewers to track the changes more efficiently, ensuring a smoother and more comprehensible review.

In Table 2 and other related tables, I suggest that the information be updated to better reflect the intervention strategies in relation to SDT. Specifically, it should be indicated whether the strategies applied are designed to promote an approach that supports psychological needs or whether there is any form of control over them. Including this information in the tables will provide greater clarity about the intervention design and its alignment with the proposed theory.

In short, the manuscript has great potential, but to ensure that the impact of the intervention and its alignment with SDT are clearly understood, I suggest incorporating the aforementioned modifications. I believe it is important to highlight whether the intervention strategies favour a supportive or controlling approach to psychological needs, as this distinction may influence the interpretation of the results and the applicability of the intervention to other contexts.

**Do you want your identity to be public for this peer review?** For information about this choice, including consent withdrawal, please see our Privacy Policy

Reviewer #1: No

Reviewer #3: No

---

## [Author Response · Author response to Decision Letter 2]

20 Oct 2025

Comment #1

Clarification on support and control strategies in SDT

One of the most relevant points in this study is the use of SDT-based strategies. However, I would like to suggest clarifying whether the autonomy, competence, and relatedness strategies implemented in the intervention are primarily aimed at supporting these psychological needs or controlling them. According to SDT, these needs can be met in ways that either support or control individuals' motivation, and it is important to highlight this point in the article, as it could have implications for the interpretation of the results. This distinction is fundamental to understanding how the intervention impacts participants' motivation and behaviors.

response:

Thanks for this comment. We edited Table 2 to indicate that all strategies were implemented to support the individuals' motivation.

Comment #2

Updating the tables

While I appreciate the author’s response to the reviewers' comments, I must point out that, for future occasions, it would be advisable to send a supplementary response to the reviewers' comments. The format in which the authors have responded to the comments makes it difficult to review the article, as the changes made to the manuscript are not clearly specified, nor are the exact sections that have been changed or added indicated. A more detailed and structured response would help reviewers to track the changes more efficiently, ensuring a smoother and more comprehensible review.

response:

Thanks for this suggestion. We will consider it next submissions.

Comment #3

In Table 2 and other related tables, I suggest that the information be updated to better reflect the intervention strategies in relation to SDT. Specifically, it should be indicated whether the strategies applied are designed to promote an approach that supports psychological needs or whether there is any form of control over them. Including this information in the tables will provide greater clarity about the intervention design and its alignment with the proposed theory.

response:

Seeing response to comment #1

---

## [Decision Letter · Decision Letter 2]

29 Oct 2025

Effects of a school-based intervention on 24-hour movement behaviors in adolescents: A quasi-experimental study.

PONE-D-25-37775R2

Dear Dr. Ricardo-Sejin,

We’re pleased to inform you that your manuscript has been judged scientifically suitable for publication and will be formally accepted for publication once it meets all outstanding technical requirements.

Kind regards,

Henri Tilga, PhD

Academic Editor

PLOS ONE

Additional Editor Comments (optional):

Reviewers' comments:

Reviewer's Responses to Questions

**Comments to the Author**

Reviewer #3: All comments have been addressed

2. Is the manuscript technically sound, and do the data support the conclusions?

Reviewer #3: Yes

3. Has the statistical analysis been performed appropriately and rigorously?

Reviewer #3: Yes

4. Have the authors made all data underlying the findings in their manuscript fully available?

Reviewer #3: Yes

5. Is the manuscript presented in an intelligible fashion and written in standard English?

Reviewer #3: Yes

Reviewer #3: Dear authors,

I have reviewed the article entitled ‘Effects of a school-based intervention on 24-hour movement behaviours in adolescents: A quasi-experimental study’ and am satisfied with the work done. I have no further input or additional comments, and I consider the article ready for publication.

Congratulations on your excellent work.

Yours sincerely,

**Do you want your identity to be public for this peer review?** For information about this choice, including consent withdrawal, please see our Privacy Policy

Reviewer #3: No

---

## [Editor Report · Acceptance letter]

PONE-D-25-37775R2

PLOS ONE

Dear Dr. Ricardo-Sejin,

I'm pleased to inform you that your manuscript has been deemed suitable for publication in PLOS ONE. Congratulations! Your manuscript is now being handed over to our production team.

Kind regards,

on behalf of

Dr. Henri Tilga

Academic Editor

PLOS ONE